# Confounds and overestimations in fake review detection: Experimentally controlling for product-ownership and data-origin

**Felix Soldner** [1,2]*, **Bennett Kleinberg**[1,3], **Shane D. Johnson**[1]

**1** Department of Security and Crime Science & Dawes Centre for Future Crime, University College London, London, United Kingdom, **2** GESIS–Leibniz Institute for the Social Sciences, Cologne, Germany, **3** Department of Methodology and Statistics, Tilburg University, Tilburg, The Netherlands

* felix.soldner@gesis.org, felix.soldner@ucl.ac.uk

**Data Availability Statement:** The data is available at: https://osf.io/29euc.

**Funding:** This work was funded by the Dawes Centre for Future Crime. The funders had no role in

## Abstract

The popularity of online shopping is steadily increasing. At the same time, fake product reviews are published widely and have the potential to affect consumer purchasing behavior. In response, previous work has developed automated methods utilizing natural language processing approaches to detect fake product reviews. However, studies vary considerably in how well they succeed in detecting deceptive reviews, and the reasons for such differences are unclear. A contributing factor may be the multitude of strategies used to collect data, introducing potential confounds which affect detection performance. Two possible confounds are data-origin (i.e., the dataset is composed of more than one source) and product ownership (i.e., reviews written by individuals who own or do not own the reviewed product). In the present study, we investigate the effect of both confounds for fake review detection. Using an experimental design, we manipulate data-origin, product ownership, review polarity, and veracity. Supervised learning analysis suggests that review veracity (60.26–69.87%) is somewhat detectable but reviews additionally confounded with product-ownership (66.19–74.17%), or with data-origin (84.44–86.94%) are easier to classify. Review veracity is most easily classified if confounded with product-ownership and data-origin combined (87.78–88.12%). These findings are moderated by review polarity. Overall, our findings suggest that detection accuracy may have been overestimated in previous studies, provide possible explanations as to why, and indicate how future studies might be designed to provide less biased estimates of detection accuracy.

## 1. Introduction

Online shopping is not new, but it is increasing in popularity as seen by the growth of companies such as Amazon and e-Bay [1–3]. Previous work shows that consumers rely heavily on product reviews posted by other people to guide their purchasing decisions [4–6]. While sensible, this has created the opportunity and market for deceptive reviews, which are currently among the most critical problems faced by online shopping platforms and those who use them

study design, data collection and analysis, decision to publish, or preparation of the manuscript.

**Competing interests:** The authors have declared that no competing interests exist.

[7, 8]. Research suggests that for a range of deception detection tasks (e.g. identifying written or verbal lies about an individual's experience, biographical facts, or any non-personal events), humans typically perform at the chance level [9, 10]. Furthermore, in the context of considering online reviews, the sheer volume of reviews [11] makes the task of deception detection implausible for all but the most diligent consumers. With this in mind, the research effort has shifted towards the use and calibration of automated approaches. For written reviews, which are the focus of this article, such approaches typically rely on text mining and supervised machine learning algorithms [12–15]. However, while the general approach is consistent, classification performance varies greatly between studies, as do the approaches to constructing the datasets used. Higher rates of performance are usually found in studies for which the review dataset [16–21] is constructed from several different sources, namely a crowdsourcing platform and an online review platform [13, 14]. High classification performances are also found in studies using data scraped or donated from a single review platform, such as Yelp or Amazon [21–24]. Lower rates of performance are typically found in studies for which data is extracted from a single source and for which greater experimental control is exercised [10, 25–27]. Why we can observe such strong differences of classification performances between studies, is unclear. However, such findings suggest that confounds associated with the construction of datasets may explain some of the variation in classification performance between studies and highlights the need for the exploration of such issues. In the current study, we will explore two possible confounds and estimate their effects on automated classification performance. In what follows, we first identify and explain the two confounds. Next, we provide an outline of how we control for them through a highly controlled data collection procedure. Lastly, we will run six analyses on subsets of the data to demonstrate the pure and combined effects of the confounds in automated veracity classification tasks.

## 1.1. Confounding factors

In an experiment, confounding variables can lead to an omitted variable bias, in which the omitted variables affect the dependent variable, and the effects are falsely attributed to the independent variables(s). In the case of the detection of fake reviews, two potential confounds might explain why some studies report higher and possibly overestimated automated classification performances than others. The first concerns the provenance of some of the data used. For example, deceptive reviews are often collected from participants recruited through crowdsourcing platforms, while "truthful" reviews are scraped from online platforms [13, 14], such as TripAdvisor, Amazon, Trustpilot, or Yelp. Creating datasets in this way is efficient but introduces a potential confound. That is, not only do the reviews differ in veracity but also their *origin*. If origin and veracity were counterbalanced so that half of the fake (and genuine) reviews were generated using each source this would be unproblematic but unfortunately in some existing studies, the two are confounded. A second potential confound concerns ownership. In existing studies, participants who write fake reviews are asked to write about products (or services) that they do not own. In contrast, in the case of the scraped reviews–assuming that they are genuine (which is also a problematic assumption)– these will be written by those who own the products (or have used the services). As such, ownership and review veracity (fake or genuine) will also be confounded.

Besides these two confounds, it is worth noting that some of the studies that have examined fake review detection have used scraped data that does not have "ground truth" labels [21–24, 28, 29]. That is, they have used data for reviews for which the veracity of the content is not known but is instead inferred through either hand-crafted rules (e.g., labeling a review as fake when multiple "elite" reviewers argue it is fake) or inferred by the platforms own filtering

system, which is non-transparent (e.g., Yelp). While utilizing such data to investigate how platforms filter reviews is helpful, studying the effects of deception without ground truth labels is problematic because any found class specific properties cannot be reliably attributed to the class label. Furthermore, any efforts to improve automated deception detection methods will be limited by algorithmically filtered data, because classification performances cannot exceed the preceding filter. Thus, ground truth labels are imperative for investigating deception detection in supervised approaches and such labels can be obtained through experimental study designs. However, previous studies collecting data experimentally [13, 14] suffer from the confounds mentioned above and an altered study design is required.

## 1.2. Confounds in fake review detection

Studies of possible confounding factors in deception detection tasks that involve reviews are scarce. In their study [30], investigated whether a machine learning classifier could disentangle the effects of two different types of deception–lies vs. fabrications. In the case of the former, participants recruited using Amazon's Mechanical Turk (AMT) were asked to write a truthful and deceptive review about an electronic product or a hotel they knew. In the case of the latter, a second group of AMT participants was asked to write deceptive reviews about the same products or hotels. However, this time they were required to do this for products or hotels they had no knowledge of, resulting in entirely fabricated reviews. [30] found that the classifier was able to differentiate between truthful reviews and fabricated ones but not particularly well. However, it could not differentiate between truthful reviews and lies–classification performance was around the chance level. These findings suggest that product ownership (measured here in terms of fabrications vs truthful reviews) is a potentially important factor in deceptive review detection.

A different study examined the ability of a classifier to differentiate truthful and deceptive reviews from Amazon [31] using the "DeRev" dataset [32]. The dataset contains fake Amazon book reviews that were identified through investigative journalism [33, 34]. Truthful reviews were selected from Amazon about other books, from famous authors such as Arthur Conan Doyle, Rudyard Kipling, Ken Follett, or Stephen King for which it was assumed that it would not make sense for someone to write fake reviews about them. A second corpus of fake reviews–written about the same books– was then generated by participants recruited through crowdsourcing to provide a comparison with the "DeRev" reviews. The authors then compared the performance of a machine learning classifier in distinguishing between different classes of reviews (e.g., crowdsourced-fake vs. Amazon-fake, crowdsourced-fake vs. Amazon-truthful). Most pertinent here was the finding that the study authors found that the crowdsourced-fake reviews differed from the Amazon-fake reviews. Both studies [30, 31] hint at the problems of confounding factors in deception detection tasks. Although [31] uses a well-designed setup to test hypotheses, book reviews were not always about the same books between classes, introducing a potential content related confound. Similarly, the machine learning classifiers used were not always cross-validated with the same data type (i.e., the training and testing data were sourced from different data subsets), complicating the interpretation of the results. In contrast, in the current study, we match product types, hold the cross-validating procedure constant across all data subsets, and extend the analyses to positive and negative reviews.

## 1.3. Aims of this paper

Confounding variables have the potential to distort the findings of studies, leading researchers to conclude that a classifier can distinguish between truthful and deceptive reviews when, in

reality, it is actually leveraging additional characteristics of the data, such as the way in which it was generated. Such confounds would mean that the real-world value of the research is limited (at best). In the current study, we employ an experimental approach to systematically manipulate these possible confounders and to measure their effects for reviews of smartphones. Specifically, we estimate the effect of *product-ownership* by collecting truthful and deceptive reviews from participants who do and do not own the products they were asked to review. To examine the effect of data-origin we also use data (for the same products) scraped from an online shopping platform. We first examine how well reviews can be differentiated by *veracity* alone (i.e., without confounds), and if classification performance changes when this is confounded with *product-ownership*, *data-origin*, or both. If *ownership* or *data-origin* do influence review content (we hypothesize that they do), reviews should be easier to differentiate when either of the two confounds is present in veracity classification, but reviews should be most easily classifiable if both confounds (*ownership*, *data-origin*) are present at the same time. Thus, our experiments allow us to assess how well a classifier can differentiate reviews based on veracity alone, and how much the confounds discussed above influence detection performances.

## 2. Data collection

### 2.1. Participants

Data were collected with Qualtrics forms (www.qualtrics.com) from participants recruited using the academic research crowd-sourcing platform Prolific (www.prolific.co). Since we wanted to collect reviews about smartphones, we wanted to make sure that all participants owned a smartphone they could write about. We achieved this by using a pre-screener question to limit the participant pool. In this case, only prolific users who use a mobile phone on a near-daily basis could take part. 1169 participants (male = 62.19%, female = 37.13%, prefer not to say = 0.007%) ranging between 18 and 65 years of age (*M* = 24.96, SD = 7.33) wrote reviews. The study was reviewed by the ethics committee of the UCL Department of Security and Crime Science and was exempted from requiring approval by the central UCL Research Ethics Committee. Participants provided written informed consent online by clicking all consent statement boxes affirming their consent before taking part in the study.

### 2.2. Experimental manipulation

We collected 1600 reviews from participants who owned the products (both truthful and deceptive reviews), and 800 from those who did not (deceptive review only). For the former, these were organized to generate 400 positive and 400 negative reviews for each of the factor (positive vs negative, deceptive vs truthful) combinations (i.e., positive-deceptive, negative-deceptive, positive-truthful, negative truthful). For the latter, participants could only write deceptive reviews and we collected 400 positive and negative of each. Reviews from owners and non-owners were collected using two Qualtrics survey forms. For both, participants were introduced to the task and asked to provide informed consent. They were then asked to indicate the current and previous brands of phones that they owned. Participants selected all applicable brands from ten choices (Samsung, Apple, Huawei, LG, Motorola Xiaomi, OnePlus, Google, Oppo, Other) without selecting a specific phone. The brands were selected based on the top selling smartphones by unit sales and market shares in 2018 and 2019 within Europe [35–38].

**2.2.1 Smartphone owners.**  For survey 1, participants were asked which phone they liked and disliked the most from their selected brands (Fig 1). The questions were presented in a randomized order, and participants had to rate their phones on a 5-point scale, replicating the Amazon product rating scale (1 star = very bad; 2 stars = bad; 3 stars = neutral; 4 stars = good;

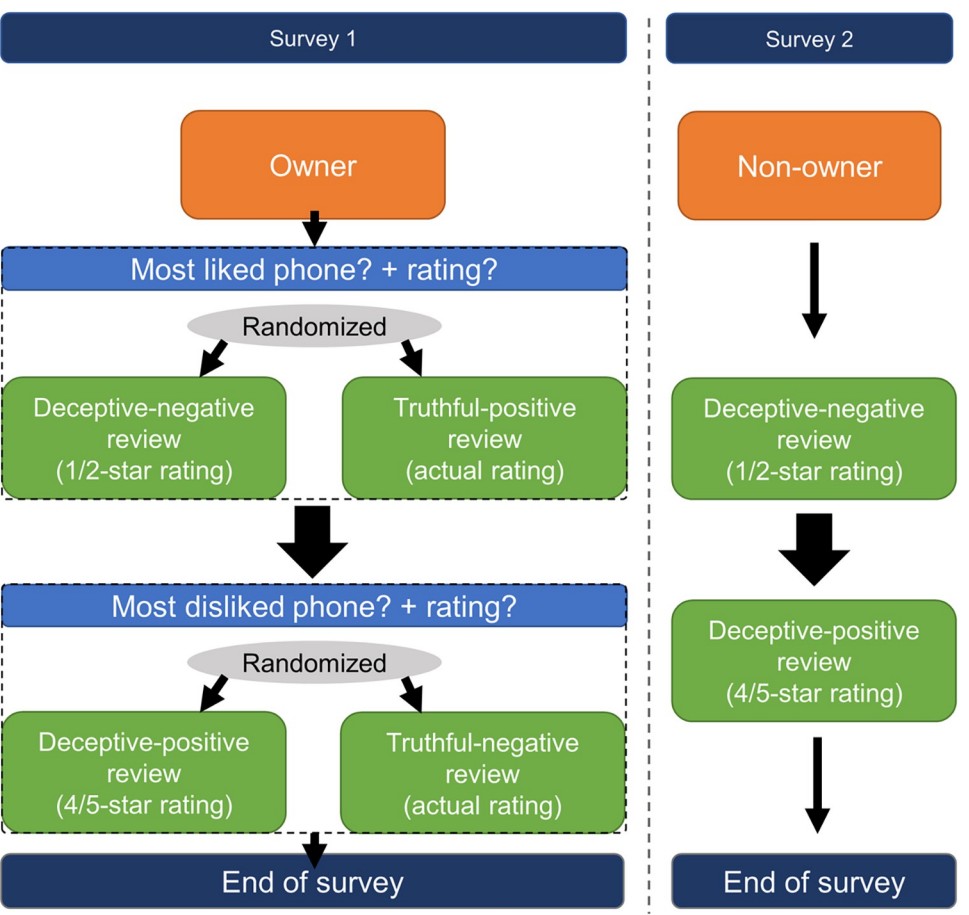

**Fig 1. Collecting procedure of reviews from Prolific participants.**

5 stars = very good). Subsequently, each participant was randomly allocated to either write a truthful or deceptive review about their most liked and their most disliked phone. The truthful review corresponded to their given phone rating. For deceptive reviews, participants were asked to write reviews that were the polar opposite of the rating they had provided. For example, for a smartphone they liked the most (or least), they were asked to write a 1- or 2-star (or 4- or 5- star) rating for that smartphone. The exact ratings (1 or 2 for a negative review, and 4 or 5 for a positive review) used for each condition (truthful or fake) were also randomized.

**2.2.2 Smartphone non-owners.** For survey 2 (Fig 1), participants were instructed to write a negative (1- or 2-star) review as well as a positive (4- or 5-star) review about two separate phones they did not own. The two randomly selected phones were selected from a list of 60 phones from the top sold phones by the top brands previously established. The brands of both phones were randomly selected from those participants who had indicated not owning them. Asking them to write about brands they did not and had not owned meant that participants could not use personal knowledge about that brand while writing the reviews. The allocation of reviews to negative and positive conditions was counterbalanced using a random number generator. Participants were allowed to perform an online search of the smartphone. Since the shortest Amazon reviews for electronic products contain around 50 characters, but most range between 100 to 150 characters [11], the minimum length of reviews participants had to write was set to 50 characters. To prevent participants from using existing reviews found online,

they were prevented from being able to copy-and-paste text into the text field in which they were required to provide their review. Also, participants were presented with an attention check in both ownership conditions after writing each review, by asking what type of review they were instructed to write (truthful or deceptive).

**2.2.3 Amazon reviews.** To obtain reviews that differed in *origin*, we collected Amazon reviews for the same phones that participants had written about. To do this, a list of all the smartphones reviewed (by owners and non-owners) was created and product links manually created for all of those that were available on Amazon and had received reviews. We used the "selectorlib" python package [39] to collect all reviews from each product link. To reduce the likelihood of collecting fake reviews, only those for which there was a verified purchase were used. The collection procedure adhered to the Amazons terms and conditions.

## 2.3 Final dataset

**2.3.1 Data filtering.** Crowdsourced reviews were excluded if participants failed the attention check (see above) or reviews were not written in English, the latter tested using the Python "langdetect" package [40]. Reviews were also removed if they did not follow the instructions. To detect the latter, we obtained the sentiment for each review using the "TextBlob" python package [41]. All reviews that had a 4 or 5-star rating but for which the sentiment score was below the neutral value of 0.00, or those with a 1,2, or 3-star rating that had a sentiment score higher than +0.50, were manually inspected. Twenty-nine reviews were removed using this procedure. Reviews with ratings of 4 or 5 stars were considered positive for the remainder of the analysis, while reviews with ratings of 1,2, or 3 star(s) were considered negative. From the 327 most-disliked phones, 158 were assigned 3-stars, but the associated reviews were sufficiently negative to be considered negative reviews. Table 1 shows the average sentiment scores (positive values indicate positive sentiment) for all review types and their associated ratings. It can be seen that the mean scores–including those for Amazon reviews–were consistent with the review ratings.

**2.3.2 Matching Amazon and Prolific reviews.** After all (Prolific and Amazon) reviews were filtered as described, they were matched according to smartphone and rating to generate complementary data sets. To do this, three review sets from Amazon were generated to mirror the three Prolific reviews sets. These were matched in terms of the smartphones reviewed and the ratings provided to reduce content-related confounds when comparing and classifying Prolific and Amazon reviews in later analyses. All Amazon reviews were considered truthful and from owners (as we only included those for which a purchase had been verified). Smartphone models that were not sold on Amazon or had only a limited number of reviews, were replaced with reviews for smartphone models from the same brand with the same ratings, resulting in 1060 replacements (see S1 Table). This was not possible for 127 reviews. For these, they were replaced with a smartphone review from a randomly selected brand with the same rating. The final dataset consisted of 4168 reviews (Table 2) which is publicly available at: https://osf.io/29euc/?view_only=d382b6f03e1444ffa83da3ea04f1a04a.

**Table 1. Average sentiment scores across reviews and their ratings.**

| | Ratings | | | | |
|---|---|---|---|---|---|
| Review type | 1 | 2 | 3 | 4 | 5 |
| Prolific [owners] | -0.18 | -0.03 | 0.11 | 0.35 | 0.42 |
| Prolific [non-owners] | -0.09 | -0.03 | x | 0.36 | 0.44 |
| Amazon [owners] | -0.05 | 0.03 | 0.10 | 0.32 | 0.43 |

**Table 2. Overview of all filtered reviews.**

|  | Truthful | | Deceptive | | Total |
|---|---|---|---|---|---|
| **Review type** | **Pos** | **Neg** | **Pos** | **Neg** | |
| Prolific owners | 384 | 327 | 302 | 348 | **1361** |
| Prolific non-owners | - | - | 352 | 371 | **723** |
| Amazon owners | 1038 | 1046 | - | - | **2084** |
| **Total** | **1422** | **1373** | **654** | **719** | **4168** |

## 3. Supervised learning analysis

N-grams, part of speech frequencies (POS), and LIWC (Linguistic Inquiry and Word Count, [42]) features were extracted from the reviews. To do this, URLs and emoticons were removed from all reviews and all characters converted to lowercase. LIWC features were then extracted. Subsequently, we removed punctuation, tokenized the text, removed stop-words, and stemmed the text data. Since the LIWC software performs text cleaning internally and to retain the measures on punctuations the LIWC features were generated first. From the cleaned data, we extracted unigrams, bigrams, and POS proportions for each text. The "WC" (word count) category from LIWC features was excluded. The python package nltk [43] was utilized for text cleaning and feature generation. Lastly, during feature preprocessing, features with a variance of 0 in each class (e.g., truthful-positive-owners, deceptive-positive-owners) were excluded to avoid any non-content related features (e.g., Amazon-specific website signs or words, such as "verified purchased") affecting the results. S2 Table provides the list of features, which were present across all analyses. In total, we removed 69,927 (99.92%) bigrams, 6,520 (94.31%) unigrams, 13 (37.14%) POS features (WRB, WP$, WP, VBG, UH, TO, RBS, PRP, POS, PDT, EX, ", $), and 4 (4.3%) LIWC features (we, sexual, filler, female).

Instead of using a pre-trained model from other published work, we decided to train and test our own classifier. So doing meant that we could ensure that all observed changes in classification performance could be attributed to our experimental manipulations as opposed to other factors (e.g., domain differences or class imbalances in the training data of other models). We tested several different classifiers, but the "Extra Trees" classifier [44] showed the best performance in most scenarios and is, therefore, reported throughout all analyses (see S3 Table for the classifier settings and S4 Table for a list of other tested classifiers). All classification models were implemented in python with the "scikit-learn" package [45]. No hyperparameter changes were made.

## 4. Results

A total of six analyses were performed to investigate how well reviews could be classified in terms of their *veracity*, *ownership*, and *data-origin* alone, as well as how strongly *ownership*, and *data-origin* affected the classification of truthful and fake reviews individually and combined. In each analysis, wherever needed, we balanced the classes by downsampling the reviews of the majority class.

### 4.1 Classification performance

Each analysis involves a binary classification task for a subset of the data (e.g., fake vs. truthful reviews, reviews from phone owners vs. non-owners, etc.), each separated into negative and positive reviews. Thus, each classification analysis contains reviews that exhibit one or more of the following: *veracity*, *ownership*, and *data-origin*. Classification performance is measured in accuracy (acc.), precision (pre.), recall, and F1, each of which is averaged across a 10-fold

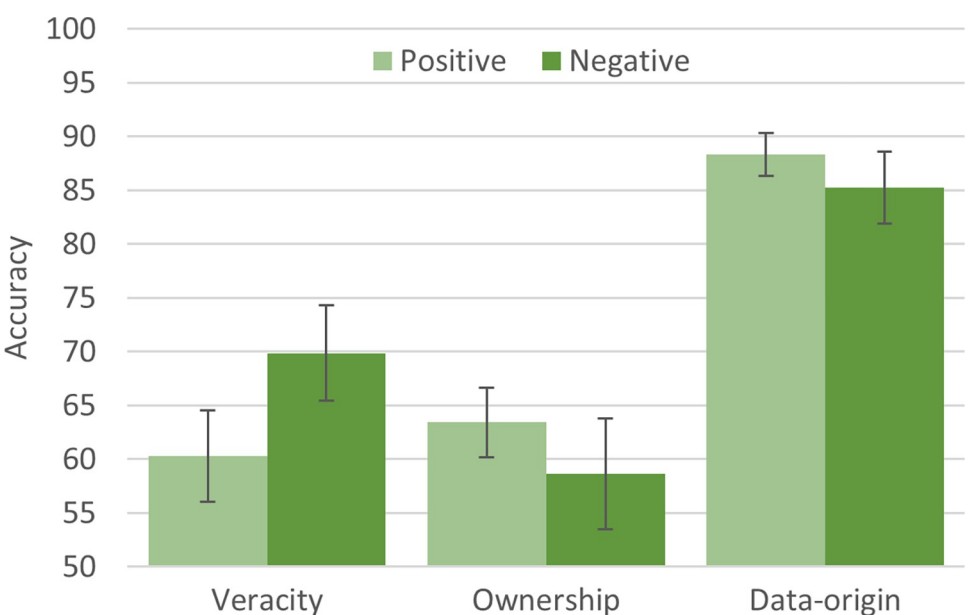

**Fig 2. Classification accuracies (with 99% CI) of analyses 1 (*veracity*), 2 (*ownership*), and 3 (*data-origin*); accuracy ranges from 50% (chance level) to 100%.**

cross-validation procedure. Since the performance metrics behaved the same between classes, we only report accuracies here (see S5 Table for a full list of all performance metrics).

**4.1.1 Pure classifications of *veracity*, *ownership*, and *data-origin*.** The first three analyses examined how well reviews can be distinguished in terms of *veracity*, *ownership*, and *data-origin*. The binary classification analyses were carried out for the following pure (i.e., removing any confounds) comparisons: **(1)** Participant [owners, fake] and participant [owners, truthful] (assessing *veracity*), **(2)** participant [non-owners, fake], and participant [owners, fake] (assessing *ownership*), and **(3)** participant [owners, truthful] and Amazon [owners, truthful] (assessing *data-origin*). Classification performance for each analysis is reported in Fig 2. The results show that the classifier found it difficult to differentiate reviews that differed solely in terms of *veracity* or *ownership* but performed better for *data-origin*. However, classification performance was different by review sentiment. Specifically, *veracity* seemed to be more easily classified if reviews were negative.

**4.1.2 Confounded classifications of *veracity*.** The last three analyses focus on the classification of *veracity* but examine how this is affected by–or confounded with–*ownership*, *data-origin*, and the two combined. The goal was to assess the strength of these factors to estimate the extent to which they (as confounders) might have affected the accuracy of classifiers in other studies. Specifically, we compared **(4)** participant [non-owners, fake] and participant [owners, truthful] (assessing *veracity confounded with ownership*), **(5)** participant [owners, fake] and Amazon [owners, truthful] (assessing *veracity confounded with data-origin*), as well as **(6)** participant [non-owners, fake] and Amazon [owners, truthful] (assessing *veracity confounded with ownership* and *data-origin*). Classification performance for each analysis (and analyses 1 for comparison) is reported in Fig 3. The results show that all confounds have a boosting effect on the *veracity* classification, but with different strengths. Compared to analysis 1 (Fig 2, assessing veracity) the veracity classification seems to be overestimated with the confound of *ownership* by 6.15–9.84%, with *data-origin* by 21.11–44.27%, and with *ownership* and *data-origin* combined by 24.89–46.23%, depending on sentiment.

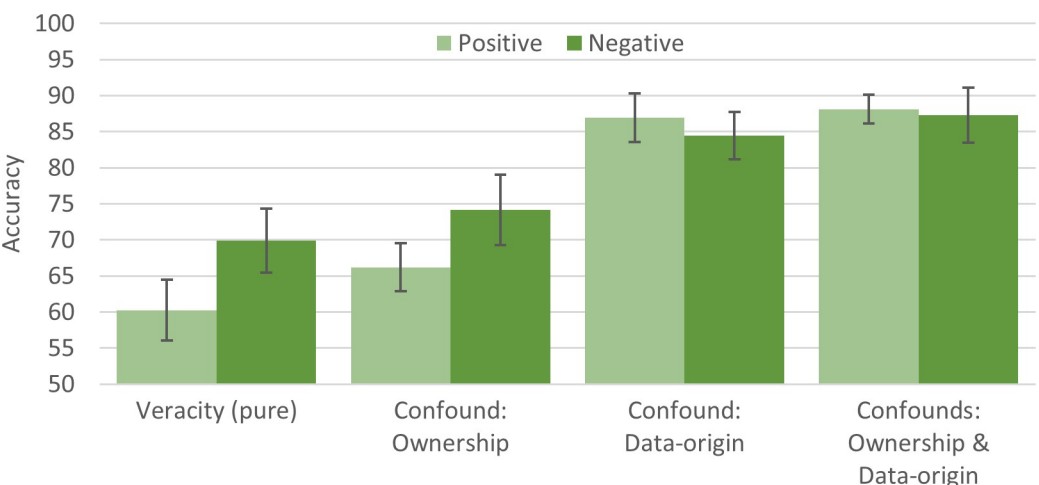

**Fig 3. Classification accuracies (with 99% CI) of analyses 1 (*veracity*), 4 (*veracity, ownership*), 5 (*veracity, data-origin*), and 6 (*veracity, ownership, data-origin*); accuracy ranges from 50% (chance level) to 100%.**

## 4.2 Linguistic properties in pure and confounded classification experiments

The linguistic properties of all analyses were examined using Bayesian hypothesis testing [46–48]. The aim was to investigate which linguistic features drive each classification. To do this, we inspected the top 5 highest Bayes Factors ($BF_{10}$) and reported features with scores of 10 or greater. A $BF_{10}$ indicates the likelihood of the data if there was a difference of occurrences in the feature between the compared classes (alternative hypothesis) relative to the null hypothesis (no difference). A $BF_{10}$ of 1 represents an equal likelihood of the null- and alternative hypothesis. Each reported feature name is tagged with one of the following indications: "POS" (part of speech), "LIWC" (Linguistic Inquiry and Word Count), "UNI" (unigram), "BI" (bigrams), to categorize its feature type. Table 3 shows the top 5 linguistic features for the pure classification experiments, while Table 4 shows the top 5 features for the classification experiments in which we introduced confounds.

**Table 3. Top 5 feature differences by $BF_{10}$ for all pure classifications; $BF_{10}>11$ for each feature; (+) = feature appears more often in truthful reviews, in reviews by smartphone non-owners, or in Prolific reviews; (-) = feature appears more often in deceptive reviews, in reviews by smartphone owners, or in Amazon reviews; see S6 Table for all feature explanations.**

| Testing | | Experiment | | |
|---|---|---|---|---|
| | | **Veracity** | **Ownership** | **Data-origin** |
| **Sentiment** | **Positive** | LIWC_social (-) | POS_CD (-) | LIWC_Comma (+) |
| | | POS_CD (+) | LIWC_see (-) | UNI_smartphon (+) |
| | | UNI_phone (-) | UNI_im (-) | UNI_camera (+) |
| | | UNI_iphon (+) | LIWC_WPS (-) | LIWC_social (-) |
| | | LIWC_WPS (+) | LIWC_i (-) | UNI_best (+) |
| | **Negative** | LIWC_focuspresent (-) | UNI_samsung (+) | UNI_smartphon (+) |
| | | LIWC_focuspast (+) | | LIWC_Comma (+) |
| | | LIWC_money (-) | | UNI_slow (+) |
| | | LIWC_Tone (+) | | LIWC_Period (+) |
| | | LIWC_conj (+) | | LIWC_ppron (-) |

**Table 4. Top 5 feature differences by Bayes Factor$_{10}$ for all confounded classifications; BF$_{10}$>188 for each feature; (+) = feature appears more often in truthful reviews; (-) = feature appears more often in deceptive reviews; see S6 Table for all feature explanations.**

| | | Experiment | | |
|---|---|---|---|---|
| Testing | | Veracity, Ownership | Veracity, Data-origin | Veracity, Ownership, Data-origin |
| Sentiment | Positive | UNI_year (+) | UNI_smartphon (-) | UNI_camera (-) |
| | | LIWC_time (+) | UNI_camera (-) | UNI_smartphon (-) |
| | | LIWC_Exclam (-) | LIWC_Authentic (+) | LIWC_article (-) |
| | | LIWC_percept (-) | LIWC_auxverb (-) | LIWC_function (-) |
| | | UNI_still (+) | POS_JJS (-) | UNI_photo (-) |
| | Negative | LIWC_focuspast (+) | LIWC_focuspast (+) | UNI_smartphon (-) |
| | | LIWC_money (-) | LIWC_Authentic (+) | LIWC_Comma (-) |
| | | LIWC_focuspresent (-) | UNI_qualiti (-) | LIWC_work (+) |
| | | UNI_buy (-) | UNI_bad (-) | UNI_slow (-) |
| | | LIWC_Exclam (-) | LIWC_social (+) | UNI_bad (-) |

## 5. Discussion

This paper investigated how product *ownership* and *data-origin* might confound interpretation of the accuracy of a machine learning classifier used to differentiate between truthful and fake reviews (*veracity*), using smartphones as a case study. To disentangle the unique contributions of each factor, we devised an experimental data collection procedure and created a dataset balanced on all factors. We used supervised learning to examine pure and stepwise confounded classification performance.

### 5.1 Classifying veracity

The supervised machine learning analyses showed that after controlling for two possible confounders (*ownership* and *data-origin*) reviews can be classified as to their pure *veracity*, but with difficulty, suggesting that detecting fake reviews may be harder than other studies have reported [13, 14]. Furthermore, at least for our data, negative reviews appear to be easier to classify (by almost 10%) than do positive ones, suggesting that the way individuals deceive differs depending on sentiment.

### 5.2 Classifying confounded *veracity*

As discussed, the two confounds tested led to an overestimation in the classification of *veracity* of between 6–46%, depending on which confound, and sentiment was involved. The combined confounding effect of *ownership* and *data-origin* (24.89–46.23% overestimation, depending on sentiment) seems to have the strongest effects, followed by *data-origin* (20.85–44.27%) and *ownership* (6.15–9.84%) alone. The ordering of these effects is the same for both positive and negative reviews. However, classification performance for positive and negative reviews differs. Specifically, negative reviews are easier to classify when *veracity* is confounded with *ownership*, but the reverse is true when *veracity* is confounded with *data-origin* or *data-origin* and *ownership* combined. Additionally, the difference in performance by review sentiment is most clear when *ownership* is involved (8.22%) than for *data-origin* (2.5%) or both combined (0.86%). The performance differences associated with the change in sentiment (e.g., veracity vs. ownership classification) further supports the idea that the veracity classification is sentiment dependent and might not be as easily generalizable.

Interestingly, when comparing the pure classification of *data-origin* to the confounded classification of *veracity* and *data-origin*, performance was almost identical. The similarity in

classification performance seems slightly counter-intuitive, as one would expect that an increase in difference would lead to an increase in performance. A possible explanation is that some of the Amazon reviews were deceptive, which would mean that they add no additional information to the classification task. The interpretability rests on the assumption that Amazon reviews are truthful, which is further discussed below in 5.5.

## 5.3 Linguistic properties

Examining the top 5 features ranked by their Bayes Factor$_{10}$ for each classification task provides insight into each class's text differences. As expected from the classification performance metrics, we observe that features are not consistent across sentiment nor between or across classes.

Both deceptive and truthful reviews highlight non-psychological or non-perceptual constructs (except LIWC_social). Given the increased usage of "phone" in deceptive positive reviews, individuals writing such reviews might reiterate what review type they are writing. Similarly, truthful review writers seem to highlight that their product originates from the brand Apple. Truthful reviews also seem to focus on the past, which might support the idea of highlighting past-owned phones. Thus, truthful negative reviews might exhibit a stronger emphasis on the ownership of the phone and when it was owned. In turn, deceptive negative reviews seem to focus more on the present and money, suggesting that the phone price might be over-emphasized in such reviews. For differences in *ownership*, we observe fewer differences, but smartphone owners seem to express their experience more in personal terms than do non-owners. However, this was only the case for positive reviews. Prolific reviews seem to follow a more factual, syntactical, and sentiment-specific style, suggesting a stronger emphasis on the product. In contrast, Amazon reviews seem to include more social processes (LIWC_social) and personal pronounces (LIWC_ppron), which could be attributed to an increased focus on services (delivery, refunds, customer support, etc.). However, the increased usage of the words "best" and "slow" in Prolific reviews could serve a similar function.

Lastly, an examination of the linguistic properties of the confounded classification experiments shows a mixed picture of classification features that appeared in the initial experiment (i.e., when no confounds were introduced) and some new ones. For example, feature differences for the classification of *veracity* and *ownership* show a new set of features for positive reviews. For the negative reviews, however, the features identified were those previously seen in the pure *veracity* classification. Since almost no differences were present for the *ownership* condition features for negative reviews, such an effect seemed expected. Interestingly, when *veracity* is confounded with *data-origin*, or with *ownership*, and *data-origin* combined, we see recurring features from the purely *data-origin* classification, which is strongest when both confounds are present. Thus, *data-origin* seems to have a strong linguistic impact, which is reflected in strong classification performance.

## 5.4 Practical implications

Previous studies that have examined the effects of supervised deception (veracity) detection suggest, that model performance does not easily transfer across domains, datasets or languages [49–52]. While domain specific language features (e.g., words specifically related to a product or service) probably contribute to the difficulties of model generalizability, other features related to the research setup or data collection procedures have also been candidate explanations for low model transferability [49]. The current study supports the idea, that the research design can have a strong effect on the classification performances in deception detection. Specifically, how the data is sourced (*data-origin*) seems to substantially effect classification

performance. Consequently, models that are trained on data with confounds, such as in *data-origin* will most likely not generalize well, and will probably perform poorly when employed on data, which is sourced differently than the models' training data. Thus, our findings indicate the importance of controlling for confounds in the training of classifiers.

## 5.5 Limitations

**5.5.1 Are verified Amazon reviews truthful?.** This study is not without limitations. Chief among these is the assumption that Amazon reviews are truthful, which rests upon the "verified purchased" seal. However, the current system used by Amazon does seem to be exploitable [53, 54]. Investigative journalists have found several potential ways to circumvent the verified purchased seal: (1) Companies send customers free products in exchange for positive reviews [55]; (2) Companies send packages to random addresses, which are registered with Amazon accounts, to obtain fake reviews [56–58]; (3) Sellers hijack reviews from other products [59]. Thus, it is plausible that some of the Amazon reviews used in this study were deceptive. Since we cannot control for reviews posted on Amazon, this is difficult to test. However, it could explain findings that showed almost equal performance when the veracity was confounded with *data-origin* (i.e., deceptive reviews from Prolific participants and truthful reviews from Amazon customers) compared to the pure *data-origin* (participants vs Amazon, both truthful reviews) test. A small part of the Amazon reviews–assumed to be truthful–could be deceptive and might lead to an increased similarity with deceptive participant reviews, making it more difficult to differentiate them. Future studies that use truthful and deceptive participant reviews could test this hypothesis. By incrementally contaminating the truthful reviews with deceptive reviews (i.e., deceptive reviews labeled as truthfully) and reporting the classification performance for each step, the changes in classification performance could be estimated. Since the differentiation should become more difficult, a drop in classification performance would be expected, which can then be tested for association with the degree of contamination. If observed, the performance drop could then serve as an indirect indicator of fake review contamination. We could then compare the percentage difference of classification performance of *veracity* (analysis 1), *data-origin* (analysis 3) and *veracity* and *data-origin* (analysis 5) to estimate the contamination of fake reviews within Amazon reviews. However, the idea that deceptive reviews from Amazon and a crowdsourcing platform are similar contradicts other findings [31]. Nonetheless, fake Amazon and fake crowdsourced reviews would only need to show some similarities or at least only be more similar to each other than fake crowdsourced to truthful Amazon reviews to have a negative effect on classification performances.

**5.5.2 Quality of Prolific reviews.** We cannot be certain that Prolific participants who wrote the smartphones reviews were honest when instructed to be. However, previous research has shown that crowdsourcing platforms produce–in most cases–high-quality data and are better suited to the collection of large amounts of text data than other traditional collection methods, such as student samples [60, 61]. Research also suggests that compared to other crowdsourcing platforms (e.g., Amazon Mechanical Turk, CloudResearch), Prolific seems to produce data of higher quality and with the most honest responses [62].

## 6. Conclusion

Through careful experimental control, we found that product *ownership* and *data-origin* do confound fake review detection. This may have resulted in the overestimation of model performance in detecting veracity in previous work. In particular, *data-origin* seems to boost classification performance, and this could easily be misattributed to the classification of *veracity* alone. Our findings suggest an overestimation of 24.89–46.23% when data is sourced from

different platforms. Consequently, more effort and experimental control are necessary to create datasets when investigating complex concepts such as deception.

## Supporting information

**S1 Table. Amazon review replacements.**
(PDF)

**S2 Table. All features used in the classification experiments.**
(PDF)

**S3 Table. Extra Trees classifier settings.**
(PDF)

**S4 Table. Other tested classifiers.**
(PDF)

**S5 Table. All classification performance metrics across all experiments.**
(PDF)

**S6 Table. Explanations of all feature names.**
(PDF)

## Author Contributions

**Conceptualization:** Felix Soldner, Bennett Kleinberg, Shane D. Johnson.

**Data curation:** Felix Soldner.

**Formal analysis:** Felix Soldner.

**Investigation:** Felix Soldner.

**Methodology:** Felix Soldner, Bennett Kleinberg, Shane D. Johnson.

**Project administration:** Felix Soldner, Bennett Kleinberg, Shane D. Johnson.

**Resources:** Felix Soldner, Bennett Kleinberg.

**Supervision:** Bennett Kleinberg, Shane D. Johnson.

**Validation:** Felix Soldner.

**Visualization:** Felix Soldner.

**Writing – original draft:** Felix Soldner.

**Writing – review & editing:** Felix Soldner, Bennett Kleinberg, Shane D. Johnson.

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
