## [Decision Letter · Decision Letter 0]

24 Feb 2022

PONE-D-21-36649Confounds and Overestimations in Fake Review Detection: Experimentally Controlling for Product-Ownership and Data-OriginPLOS ONE

Dear Dr. Soldner,

Thank you for submitting your manuscript to PLOS ONE. After careful consideration, we feel that it has merit but does not fully meet PLOS ONE’s publication criteria as it currently stands. Therefore, we invite you to submit a revised version of the manuscript that addresses the points raised during the review process. In preparing this revision I suggest to carefully follow the thorough comments provided by the two reviewers. In particular, more effort should be devoted at critically analyzing the large body of related papers. In doing this, particular care should be given at emphasizing differences with respect to the 2020 paper by Fornaciari et al. Then, more details, explanations and experiments are also needed, as highlighted by both reviewers.

We look forward to receiving your revised manuscript.

Kind regards,

Stefano Cresci

Academic Editor

PLOS ONE

Journal Requirements:

2. Please include a complete ethics statement in the Methods section, including information on how participants were recruited, the name of the IRB and the approval number, and whether they approved the study or waived the need for approval. Please also clarify whether the participant provided consent, and if so, how, or whether the IRB waived the need for consent." 2) "Please state in the Methods section whether or not the data collection from online sources complied with the terms and conditions of the services used.

3. We note you have included a table to which you do not refer in the text of your manuscript. Please ensure that you refer to Tables 3, 4 and 5 in your text; if accepted, production will need this reference to link the reader to the Table

Reviewers' comments:

Reviewer's Responses to Questions

**Comments to the Author**

1. Is the manuscript technically sound, and do the data support the conclusions?

Reviewer #1: Partly

Reviewer #2: Yes

2. Has the statistical analysis been performed appropriately and rigorously? 

Reviewer #1: N/A

Reviewer #2: Yes

3. Have the authors made all data underlying the findings in their manuscript fully available?

Reviewer #1: Yes

Reviewer #2: Yes

4. Is the manuscript presented in an intelligible fashion and written in standard English?

Reviewer #1: Yes

Reviewer #2: Yes

5. Review Comments to the Author

Reviewer #1: The manuscript considers possible confounding factors that may alter the results of a fake review classification task. Specifically, it considers two properties: the origin of the reviewer (e.g., social platform or crowdsourcing?) and whether or not the reviewer owns the product being reviewed.

In order to understand if the classifier is influenced by these two features, in addition to the veracity of the review, the authors build a review dataset that can be filtered according to three different aspects: the veracity of the review, the data origin of the reviewer, the ownership of the product by the reviewer.

The paper poses a very interesting problem, that is, how to build a suitable and non-biased dataset for a classification task.

A first point that puzzles me is that probably the dataset that the authors have built is also biased... certainly not because of the authors.

First of all, as they themselves acknowledge, to consider Amazon reviews as truthful is to deny the very existence of the problem of fake reviews... And the authors are well aware that even certified purchases can be manipulated, as they point out in the discussion.

Also, how can we be sure that Prolific reviewers aren't lying? Unfortunately, that cannot be verified either, or at least I think so. I have tried registering on Prolific (thanks for introducing me to it, very interesting!), but they do not currently need annotators, so I do not know if they simply trust what an annotator says during registration, and if not, how do they verify the annotators' supposed characteristics?

Anyway, I understand that this is really a big problem to solve. I think it could be useful to enrich the discussion to consider the problem of reliability of crowdsourcing platforms as well.

There is one assessment that seems to me a bit too exclusive, about data origin. Quoting from the text:

....`The first concerns the provenance of the data used. For example, deceptive reviews are often collected from participants recruited through crowdsourcing platforms, while truthful reviews are scraped from online platforms.’

This is true for the cited articles, the famous articles by Ott et al., but the literature also shows many examples of datasets built with reviewers with same data origin, e.g., with tags regarding veracity released by the same platforms (there are some work based on Yelp’s release of tags). Then, the training is done on that dataset, and then the trained classifier lis launched on another unknown dataset, where, maybe, the reviewer is of another origin (say, trained on Yelp and used on Amazon, for example..). I think the paper deserves a slightly more robust RW part. The following paper mentions different approaches in the Literature Review section. So maybe authors can take a cue:

Fazzolari et al.:

Experience: Improving Opinion Spam Detection by Cumulative Relative Frequency Distribution. ACM J. Data Inf. Qual. 13(1): 4:1-4:16 (2021)

We now come to performances. Unfortunately, I cannot interpret the results -or better, I cannot interpret the authors interpretation of the results :), so I need more information.

I quote:

….`leading researchers to conclude that a classifier can distinguish between truthful and fake reviews when, in reality, it is actually distinguishing between other characteristics of the data.’

Good, but the point is: do these additional features that the dataset has, which the authors call confounding factors, help or not to find fake reviews? If I understand correctly, yes! (Am I wrong, is there something I'm not clear on?). Because if I look at the performances, and read about overestimates, I don't understand what this overestimation is). Can you give me an example where, in a certain scenario, confounding factors lead to classify fake reviews as true or vice versa? Is this the issue?

Minor: the abstract is not very understandable, or rather, the first time I read it I understood very few.. Probably the authors can try rephrasing to introduce the reader more gently to the paper subject.

In conclusion, I believe that it should be better clarified

1) if and how much bias could have been introduced while building the dataset.

2) why the authors talk about `confounding’ factors, do these factors help or not to find fake reviews?

3) in my opinion, the literature review on fake review detection should be expanded, with emphasis on datasets building.

Reviewer #2: The paper focuses on an intriguing question that the literature systematically underestimates. It also reads well and smoothly up to the end, even considering the quite intertwined set of experiments.

The results appear to be sound and supported by the plots. However, as reference 17 shows, the results are not completely unexpected. Moreover, there are several issues that the authors should consider addressing to improve the paper.

Firstly, the model used in the experiments is unclear and unexplained (maybe overlooked). Since the literature is flourishing with models of classifiers that excel (as often declared by authors) in detecting fake reviews, the authors could have explained their choice not to adopt available classifiers and build their own classifier. Similarly, the details of the used classifier are pretty limited (a Section with a total of 11 lines): the reader would expect at least the list of the final employed features and some additional elements to better figure out the mechanism chosen to process the dataset. Considering that the whole paper is centered on the effects of confounds on classifier performances, the reader should expect better awareness and care of such details. While the focus on the data is crucial, the reliability and quality of the classifier building are also critical and should not be neglected.

Finally, there are also some disregard, maybe naive, observations about why the obtained results are strikingly evident when comparing crowdsourced reviews with Amazon's ones. In Amazon's reviews, it is somewhat expected to encounter considerations to elements out of the reviewed product itself, namely references to the shipping time, delivery status, customer service experiences, returns, refunds, refurbished products, and similar. While the authors in Section 4 refer to some form of avoidance of non-content related features, there is no reference to such (expected?) bias in the text of Amazon's reviews. A similar statement could be made when dealing with the owner and not-owner experiment: reference to relative first-person pronouns and adjectives could be expected to be higher in the owner group. This and the above observations could appear to be easy biases that the authors could further explore to explain their thesis better.

Minor typos:

-the value in the last column of Amazon owners row of table 2 appears to be a copy-n-paste typo.

-the double rounded arrow of Figure 1 is misleading: what does it mean? Maybe another label could help.

6. PLOS authors have the option to publish the peer review history of their article (what does this mean?). If published, this will include your full peer review and any attached files.

Reviewer #1: No

Reviewer #2: No

---

## [Author Response · Author response to Decision Letter 0]

28 Apr 2022

Please see the attached document “Response to reviewers” for the detailed responses.

---

## [Decision Letter · Decision Letter 1]

29 Sep 2022

PONE-D-21-36649R1Confounds and Overestimations in Fake Review Detection: Experimentally Controlling for Product-Ownership and Data-OriginPLOS ONE

Dear Dr. Soldner,

Thank you for submitting your manuscript to PLOS ONE. After careful consideration, we feel that it has merit but does not fully meet PLOS ONE’s publication criteria as it currently stands. Therefore, we invite you to submit a revised version of the manuscript that addresses the points raised during the review process. In particular, a reviewer asked for some additional clarifications and linguistic edits.

We look forward to receiving your revised manuscript.

Kind regards,

Stefano Cresci

Academic Editor

PLOS ONE

Journal Requirements:

1. Consent Information not present At PRTC, please send back with the following note.

"Please amend your current ethics statement to address the following concerns:

a) Did participants provide their written or verbal informed consent to participate in this study?

b) If consent was verbal, please explain i) why written consent was not obtained, ii) how you documented participant consent, and iii) whether the ethics committees/IRB approved this consent procedure."

Reviewers' comments:

Reviewer's Responses to Questions

**Comments to the Author**

1. If the authors have adequately addressed your comments raised in a previous round of review and you feel that this manuscript is now acceptable for publication, you may indicate that here to bypass the “Comments to the Author” section, enter your conflict of interest statement in the “Confidential to Editor” section, and submit your "Accept" recommendation.

Reviewer #1: (No Response)

2. Is the manuscript technically sound, and do the data support the conclusions?

Reviewer #1: Yes

3. Has the statistical analysis been performed appropriately and rigorously? 

Reviewer #1: N/A

4. Have the authors made all data underlying the findings in their manuscript fully available?

Reviewer #1: Yes

5. Is the manuscript presented in an intelligible fashion and written in standard English?

Reviewer #1: Yes

6. Review Comments to the Author

Reviewer #1: I thank the authors for considering my requests and making the purpose of the article clearer.

A couple of notes:

1) In articles that exploit data from the platforms themselves, like Yelp's, I am not 100% sure that the data is actually donated by the platform...I rather believe that the content of the site was pulled down by crawlers or scrapers, and that a part of reviews was defined as false because of hand crafted rules, like user alerts, while a part is the so called `filtered reviews', the ones that Yelp decided not to make public, because fake according to its algorithm, but that by scraping could be found...

thus, I'd ask the authors to possibly check the passage inserted in Section 1.1, to make sure the claims are correct.

2) As mentioned, I am very pleased that abstract, intro, conclusions... are now smoother and the article goal clearer.

Many times, however, the concept of confounding factors is repeated, and how they may have a bearing on the classifier's outcome. For my personal taste, I would avoid too much repetition of the concept. I leave it up to the editor to decide on this aspect, which is more about style.

7. PLOS authors have the option to publish the peer review history of their article (what does this mean?). If published, this will include your full peer review and any attached files.

Reviewer #1: No

---

## [Author Response · Author response to Decision Letter 1]

11 Oct 2022

Please see the attached document “Response to reviewers” and "Revised Manuscript with Track Changes" for the detailed responses and changes made.

---

## [Decision Letter · Decision Letter 2]

6 Nov 2022

Confounds and Overestimations in Fake Review Detection: Experimentally Controlling for Product-Ownership and Data-Origin

PONE-D-21-36649R2

Dear Dr. Soldner,

We’re pleased to inform you that your manuscript has been judged scientifically suitable for publication and will be formally accepted for publication once it meets all outstanding technical requirements.

Kind regards,

Nguyen Quoc Khanh Le

Academic Editor

PLOS ONE

Additional Editor Comments (optional):

Reviewers' comments:

Reviewer's Responses to Questions

**Comments to the Author**

1. If the authors have adequately addressed your comments raised in a previous round of review and you feel that this manuscript is now acceptable for publication, you may indicate that here to bypass the “Comments to the Author” section, enter your conflict of interest statement in the “Confidential to Editor” section, and submit your "Accept" recommendation.

Reviewer #1: All comments have been addressed

2. Is the manuscript technically sound, and do the data support the conclusions?

Reviewer #1: (No Response)

3. Has the statistical analysis been performed appropriately and rigorously? 

Reviewer #1: (No Response)

4. Have the authors made all data underlying the findings in their manuscript fully available?

Reviewer #1: (No Response)

5. Is the manuscript presented in an intelligible fashion and written in standard English?

Reviewer #1: (No Response)

6. Review Comments to the Author

Reviewer #1: (No Response)

7. PLOS authors have the option to publish the peer review history of their article (what does this mean?). If published, this will include your full peer review and any attached files.

Reviewer #1: No

---

## [Editor Report · Acceptance letter]

10 Nov 2022

PONE-D-21-36649R2 

Confounds and Overestimations in Fake Review Detection: Experimentally Controlling for Product-Ownership and Data-Origin 

Dear Dr. Soldner:

I'm pleased to inform you that your manuscript has been deemed suitable for publication in PLOS ONE. Congratulations! Your manuscript is now with our production department. 

Kind regards, 

on behalf of

Dr. Nguyen Quoc Khanh Le 

Academic Editor

PLOS ONE